# Learning from Demonstration with Implicit Nonlinear Dynamics Models

## Abstract

Learning from Demonstration (LfD) is a useful paradigm for training policies that solve tasks involving complex motions, such as those encountered in robotic manipulation. In practice, the successful application of LfD requires overcoming error accumulation during policy execution, i.e. the problem of drift due to errors compounding over time and the consequent out-of-distribution behaviours. Existing works seek to address this problem through scaling data collection, correcting policy errors with a human-in-the-loop, temporally ensembling policy predictions or through learning a dynamical system model with convergence guarantees. In this work, we propose and validate an alternative approach to overcoming this issue. Inspired by reservoir computing, we develop a recurrent neural network layer that includes a fixed nonlinear dynamical system with tunable dynamical properties for modelling temporal dynamics. We validate the efficacy of our neural network layer on the task of reproducing human handwriting motions using the LASA Human Handwriting Dataset. Through empirical experiments we demonstrate that incorporating our layer into existing neural network architectures addresses the issue of compounding errors in LfD. Furthermore, we perform a comparative evaluation against existing approaches including a temporal ensemble of policy predictions and an Echo State Network (ESN) implementation. We find that our approach yields greater policy precision and robustness on the handwriting task while also generalising to multiple dynamics regimes and maintaining competitive latency scores.

## 1 Introduction

Learning from demonstration is a valuable paradigm for training policies that imitate motions demonstrated by an expert. The practical value of this paradigm is especially evident in the field of robotics where it has been applied to learn policies for a wide range of real-world tasks such as stacking kitchen shelves Team et al. (2024), slotting batteries into a remote control device Zhao et al. (2023), performing surgical incisions Straižys et al. (2023) and knot tying Kim et al. (2024). In spite of this and other recent progress in LfD for robot automation Brohan et al. (2022); Chi et al. (2023); policies and the motions they generate often struggle to meet the: (a) precision, (b) latency and (c) generalisation requirements of real-world tasks [1]. Existing approaches seek to address this through scaling data collection Padalkar et al. (2023); Khazatsky et al. (2024), applying data augmentations Zhou et al. (2023); Ankile et al. (2024), ensembling model predictions Zhao et al. (2023) and explicitly enforcing policy optimisation constraints that guarantee convergence Khansari-Zadeh & Billard (2011). The efficacy of each of these approaches is however limited by the policy parameterisation itself (e.g. the precision of ensembled policy predictions relies on the precision of individual policy predictions).

One set of approaches view LfD through the lens of learning policies that approximate dynamical systems Billard et al. (2022). This characterisation of LfD has resulted in policy parameterisations and optimisation procedures that combine analysis from dynamical system theory Strogatz (2018) with tools from machine learning Bishop & Nasrabadi (2006) to create reactive policies with con-

---

[1] where precision refers to satisfying position and velocity tolerances, latency refers to the responsiveness of the policy to changes in the environment and generalisation refers to the ability to successfully adapt to changing task environments.

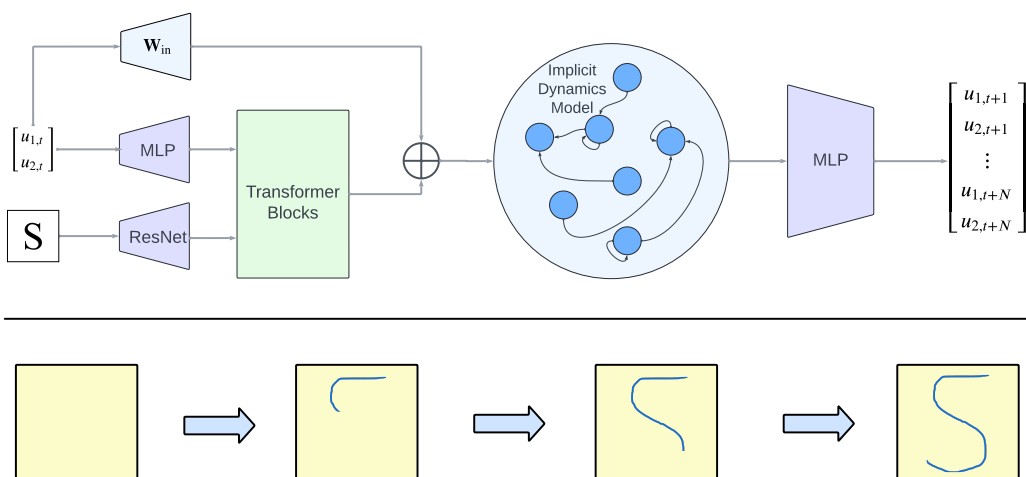

Figure 1: **Top:** A high-level overview of our architecture for the multi-task human handwriting problem. The current pen coordinates represented by $[u_{1,t}, u_{2,t}]$ are mapped to a learnable and non-learnable embeddings using an MLP and fixed linear map $\mathbf{W}_{\text{in}}$ respectively. An image of the character to be drawn is also mapped to a learnt embedding using a ResNet layer. All learnt embeddings are concatenated and passed through a sequence of self-attention blocks. The resulting embeddings are added to the non-learnable embedding of the pen state and then passed as input to the implicit nonlinear dynamics model which generates a new dynamical system state which is mapped to predicted pen coordinates. **Bottom:** Visualisation of a predicted trajectory for the character "S".

vergence guarantees Khansari-Zadeh & Billard (2011). The strengths of this approach are demonstrated in policies capable of robotics applications as demanding as catching objects in flight Kim et al. (2014). This level of performance does come at the cost of policy generalisation as task specific tuning and optimisation constraints are required in order to ensure convergence. In addition, combining multiple policies representing dynamical systems remains non-trivial as seen in work by Shukla & Billard (2012) where a robot learns different grasping dynamics based on its initial state. Generalising this approach to multi-task settings with a large variety of tasks remains an open problem.

In contrast, progress in end-to-end imitation learning with deep neural networks has resulted in policies that can generalise zero-shot across a range of language-annotated robot manipulation tasks Jang et al. (2022); Brohan et al. (2022); Team et al. (2024). However, these policies can suffer from the issue of compounding errors and the motions they generate may not accurately reflect the dynamics of expert demonstrations. Furthermore, existing methods that seek to address these shortcomings result in tradeoffs between how accurately the policy models the dynamics of expert demonstrations, the robustness of the policy to perturbations/noise and the smoothness of motions (i.e. jerk in the trajectory) Zhao et al. (2023). The disparity between the performance of explicit dynamical system learning with specialised policy parameterisations and end-to-end imitation learning with deep neural networks motivates our work to develop a policy parameterisation that combines insights from techniques for the analysis of dynamical system properties with the generalisation benefits of modern deep neural network architectures.

Reservoir Computing (RC) is the computational paradigm we draw inspiration from when designing our policy parameterisation. This paradigm has traditionally been used to model complex dynamical systems Yan et al. (2024); Kong et al. (2024), however, unlike the explicit dynamical system learning discussed in the previous paragraph, RC relies on the properties of so-called reservoirs that implicitly model the dynamics of the target dynamical system. We argue that this paradigm holds great potential as the sequences of representations it generates can be understood in terms of dynamical system properties that can be steered/optimised. Crucially, the dynamical system properties of the reservoir constitute temporal inductive biases that are not easily modelled in existing neural network architectures. For instance, the *echo state property* Yildiz et al. (2012) can be used to enforce the condition that sequences of representations generated by the dynamical system are asymptotically determined by the input data stream Sec A.1; hence the network dynamics naturally model the tem-

poral relationship between model inputs. This is in contrast to existing architectures that rely on passing a history of observations per prediction. Surprisingly, the applications of RC to learn policies that imitate expert demonstrations is relatively scarce, especially in the field of robotics Joshi & Maass (2004). In this paper, we explore the combination of concepts from RC with advancements in training deep neural networks for learning to imitate human demonstrated motions. In particular, we introduce a new neural network layer containing a fixed nonlinear dynamical system whose construction is inspired by RC literature on Echo State Networks (ESNs) Jaeger & Haas (2004), we refer to this as an Echo State Layer (ESL). We validate our contribution on the task of reproducing human handwriting motions using the openly available Human Handwriting Dataset from LASA Khansari-Zadeh & Billard (2011). This task of reproducing human handwriting motion is representative of the core learning issues associated with a large class of robot motion control problems, hence it has been studied before in the literature on representation learning Graves (2013); Khansari-Zadeh & Billard (2011); Lake et al. (2015). While it is of interest to roboticists, the framing of the problem in this present paper applies more broadly to sequential decision making and requires modest computational and hardware resources in comparison to more elaborate robotics experiments. Therefore, we hope this makes the present work easier to reproduce by others in the research community.

Our contributions are as follows:

1. We introduce a novel neural network layer called an Echo State Layer (ESL) that incorporates a nonlinear dynamics model satisfying the echo state property. This helps address the problem of compounding errors in LfD. This dynamics model incorporates learnable input embeddings, hence it can readily be incorporated into intermediate layers of existing neural network architectures.

2. We validate the performance of architectures incorporating the ESL layer through experiments in learning human handwriting motions. For this purpose, we leverage the Human Handwriting Dataset from LASA, and perform evaluation of precision, latency and generalisation.

3. We outline a roadmap for future work aimed at advancing neural network architecture design through incorporating concepts from reservoir computing and dynamical systems theory.

4. We release a JAX/FLAX library for creating neural networks that combine fixed nonlinear dynamics with neural network layers implemented in the FLAX ecosystem.

## 2 RELATED WORK

### 2.0.1 LEARNING FROM DEMONSTRATION AS LEARNING DYNAMICAL SYSTEMS

Learning from demonstration can be formalised through the lens of learning a dynamical system Billard et al. (2022). A canonical example is given by an autonomous first-order ODE of the form:

$$
\begin{aligned}
f &: \mathbb{R}^N \to \mathbb{R}^N \\
\dot{x} &= f(x)
\end{aligned}
\tag{1}
$$

where $x$ represents the system state and $\dot{x}$ the evolution of the system state. We can seek to learn an approximation of the true underlying system by parameterising a policy $\pi_\theta \approx f$ and learning parameters $\theta$ from a set of $M$ expert demonstrations $\{\mathbf{X}, \dot{\mathbf{X}}\} = \{X^m, \dot{X}^m\}_{m=1}^M$ of the task being solved. A benefit of this formulation is the rich dynamical systems theory it builds upon Strogatz (2018), enabling the design of dynamical systems with convergence guarantees Khansari-Zadeh & Billard (2011); Kim et al. (2014); Shukla & Billard (2012). For the task of reproducing human handwriting motions, Khansari-Zadeh & Billard (2011) design a learning algorithm called Stable Estimator of Dynamical Systems (SEDS) that leverages the constrained optimisation of the parameters of a Gaussian Mixture Regression (GMR) model with the constraints ensuring global asymptotic stability of the policy, hence addressing the issue of compounding errors due to distributions mismatch. A crucial limitation of this approach however is the expressiveness and generality of the policy parameterisation and requirements for formulating optimisation constraints. For instance the optimisation constraints require the fixed point of the dynamical system to be known and the policy

parameterisation requires the hyperparameter for the number of Gaussians to be optimised for the given task. Furthermore, results from applying SEDS to the LASA Human Handwriting Dataset demonstrate predicted dynamics that generate trajectories which don't perfectly model the trajectories of all characters in the demonstration data such as the "S" and "W" characters. Our work takes inspiration from the robustness of the dynamical systems modelling approach but chooses to implicitly rather than explicitly model system dynamics.

### 2.0.2 LEARNING FROM DEMONSTRATION WITH DEEP NEURAL NETWORKS

A purely statistical viewpoint of LfD seeks to learn the parameters $\theta$ of a policy $\pi_\theta$ that models a distribution over actions from expert demonstrations $\mathcal{D}$ while simultaneously demonstrating the ability to generalise beyond the demonstration dataset. The dominant policy parameterisation for LfD from this perspective is that of Deep Neural Networks (DNNs) which can coarsely be classified into three classes feedforward, autoregressive and recurrent.

Feedforward DNN architectures don't explicitly maintain temporal dynamics; their predictions solely depend on their inputs at a given instance. Inspite of the apparent limitations of feedforward architectures in modelling sequential data they can be efficacious with their success often hinging on sufficient data coverage and/or the learning of generalisable representations. In LfD for robot manipulation the ability of feedforward architectures to generalise across tasks was demonstrated by Jang et al. (2022) where their policy makes predictions conditioned on representations of language and video demonstrations, hence demonstrating the power of generalisable representations. Building upon the generalization performance seen in Jang et al. (2022), the authors of the RT-X class of models Brohan et al. (2022); Padalkar et al. (2023) incorporate the transformer attention mechanism Vaswani et al. (2017) into their architecture while simultaneously scaling the amount of data they train on yielding further improvements in policy performance across robot manipulation tasks. Subsequent advances in feedforward architectures for robot manipulation have incorporated flexible attention mechanisms Team et al. (2024), output transformations Chi et al. (2023) and techniques of chunking and smoothing predictions Zhao et al. (2023). In spite of progress in feedforward architectures for LfD, the lack of temporal network dynamics remains a potentially significant oversight and it is worth questioning whether this oversight is costly to longer-term progress.

Unlike purely feedforward architectures, autoregressive architectures simulate temporal dynamics by autoregressively passing predictions to the architecture in order to condition future predictions on a history. Having been predominantly developed for language modelling Vaswani et al. (2017); Devlin (2018), autoregressive architectures have also been used in LfD for robot motion control Reed et al. (2022); Lee et al. (2021). While modelling tasks with a temporal component is feasible using autoregressive architectures it is worth considering the limitations of the architecture, especially in tasks with complex dynamics. A key limitation of the autoregressive architecture is the latency of model predictions that results from autoregression. Furthermore, the memory requirements of such models grow with the sequence length being modelled, depending on the specific architecture Vaswani et al. (2017) the growth may be $\mathcal{O}(I^2)$ for input length $I$. Clever engineering can be employed to help address these issues Munkhdalai et al. (2024); Moritz et al. (2020), yet again it is worth questioning what constraints these limitations impose on the architecture and the consequences they have for longer-term progress.

Recurrent neural network (RNN) architectures explicitly model temporal dynamics and are naturally suited towards modelling sequential data by design. RNN architectures have previously been applied to the task of modelling human handwriting motions, most notably Graves (2013) demonstrates the ability of a deep LSTM model Hochreiter & Schmidhuber (1997) to learn to generate handwriting from human pen traces. While existing works have demonstrated the efficacy of RNN architectures in sequence modelling, these architectures struggle to scale due to multiple issues including the vanishing gradient problem Pascanu et al. (2013), challenges in modelling long-range dependencies Hochreiter & Schmidhuber (1997) and training efficiency. Our proposed neural network layer falls under the class of recurrent neural network layers, in contrast to existing recurrent neural network layers that rely on learning hidden state embeddings our approach relies on sequences of representations generated by a fixed nonlinear dynamical system.

### 2.0.3 Reservoir Computing

Reservoir computing is a computational paradigm that relies on learning from high-dimensional representations generated by dynamical systems commonly referred to as *reservoirs* Yan et al. (2024). A reservoir can either be physical Nakajima et al. (2021) or virtual Jaeger & Haas (2004); we focus on the latter as our proposed architecture digitally simulates a reservoir. A Reservoir Computer (RC) is formally defined in terms of a coupled system of equations; following the convention outlined in Yan et al. (2024) that describes the reservoir dynamics and an output mapping as follows:

$$\begin{cases} \Delta x = F(x; u; p), \\ y = G(x; u; q). \end{cases} \tag{2}$$

where $u \in \mathbb{R}^{N_{\text{in}}}$ is the reservoir input, $x \in \mathbb{R}^{N_{\text{dynamics}}}$ the reservoir's internal state and $y \in \mathbb{R}^{N_{\text{out}}}$ the output prediction. The parameters of the system are represented by $p$ and $q$ respectively where $p$ is fixed and corresponds to the reservoir dynamics while q is learnable and associated with an output mapping commonly referred to as the readout. Reservoir computing variants have traditionally been used to model temporal data, demonstrating state-of-the-art performance in the modelling of nonlinear dynamical systems Kong et al. (2024). Common instantiations of reservoir computers include Echo State Networks (ESNs) Jaeger & Haas (2004) and Liquid State Machines (LSMs) Maass et al. (2002). The dynamical systems or reservoirs we construct within our proposed neural network layer are inspired by ESNs and hence we provide a preliminary introduction in the following paragraph.

Echo State Networks (ESNs) proposed by Jaeger & Haas (2004) are a class of reservoir computers in which the reservoir is represented by a sparsely connected weighted network that preserves the *echo state property* Yildiz et al. (2012). The reservoir is generated through first sampling an adjacency matrix $A$ to represent graph topology and subsequently a weight matrix $\mathbf{W}_{\text{dynamics}}$ representing the connection strength between nodes in the graph where disconnected nodes have zero connection weight by default. In addition to generating a graph representing the reservoir, a linear mapping $\mathbf{W}_{\text{in}}$ from input data to the dimension of the reservoir (i.e. the number of nodes in the graph) is also sampled. The reservoir weights are scaled appropriately in order to maintain the *echo state property* using the *spectral radius* $\rho$ of the weight matrix $\mathbf{W}_{\text{dynamics}}$. Both the reservoir and input mapping weights are fixed after construction and leveraged in the reservoir state dynamics equations (corresponding to $\Delta x$ in Eqn 2) as follows:

$$\begin{aligned} \tilde{\mathbf{x}}(t) &= tanh(\mathbf{W}_{\text{in}}\mathbf{u}(t) + \mathbf{W}_{\text{dynamics}}\mathbf{x}(t-1)) \\ \mathbf{x}(t) &= (1-\alpha)\mathbf{x}(t-1) + \alpha\tilde{\mathbf{x}}(t) \end{aligned} \tag{3}$$

where $\tilde{\mathbf{x}}(t), \mathbf{x}(t) \in \mathbb{R}^{N_{\text{dynamics}}}$ represent node update and state at timestep $t$ respectively, $\mathbf{u}(t) \in \mathbb{R}^{N_{\text{in}}}$ are the input data and $\alpha \in [0, 1]$ is a leak rate parameter that controls the speed of state updates. The output mapping ($G$ in Eqn 2) commonly referred to as the readout is represented by a linear map $\mathbf{W}_{\text{out}}$, resulting in the following equation for output predictions $\mathbf{y}$:

$$\mathbf{y}(t) = \mathbf{W}_{\text{out}}\mathbf{x}(t) \tag{4}$$

ESNs are traditionally optimised using ridge regression on a dataset of input, output pairs. For a more comprehensive introduction and practical guide to implementing and optimising ESNs we refer the reader to Lukoševičius (2012).

## 3 Approach

Our architecture shown in Fig 1, combines the echo state layer we introduce with traditional neural network layers. In Sec 3.1.1-3.1.2 we outline the nonlinear dynamics model that we propose in the echo state layer, highlighting the mechanism by which it incorporates learnt embeddings. After introducing this nonlinear dynamics model we discuss overall architecture parameter optimisation. Architectures incorporating echo state layers are composed of both learnable and fixed parameters, as a result, we provide details of how each is optimised, with dynamics model parameter optimisation outlined in Sec 3.2.1 and learnable neural network parameter optimisation outlined in Sec 3.2.2.

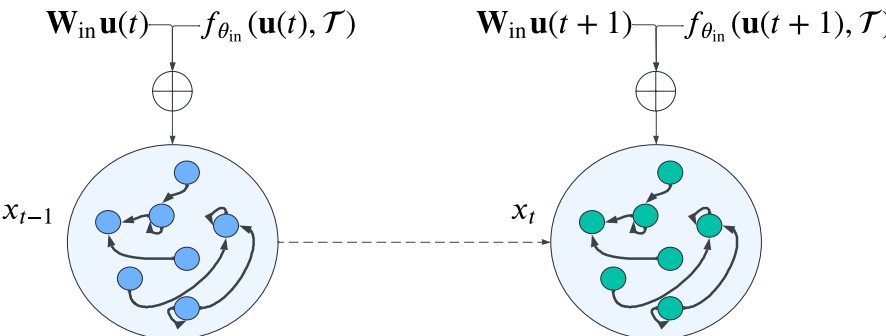

Figure 2: Our dynamics layer adds embeddings that result from fixed and learnable transformations respectively before passing them to the computational graph representing our dynamical system. Through the discrete-time forward dynamics defined in Eqn 3 we generate the next state of the dynamical system which is used to predict actions.

## 3.1 MODEL ARCHITECTURE

### 3.1.1 NONLINEAR DYNAMICS MODEL

Our discrete-time nonlinear dynamics model is inspired by the construction of a reservoir in ESNs Jaeger & Haas (2004) and the dynamics outlined in Eqn 3. In contrast to ESNs we incorporate learnable embeddings as inputs into our dynamics model making it compatible with existing neural network architectures. Similar to ESNs, the dynamics are represented by a weighted graph with specific discrete-time dynamics. We start constructing our dynamics model by sampling the topology of a network graph, in our implementation we sample node connections at random with only 1% of nodes in the graph maintaining connections, this is represented by our adjacency matrix $A$. Given the resulting sparsely connected graph topology $A$, we seek to define a weighted network graph that results in dynamics which satisfy the echo state property Yildiz et al. (2012). To accomplish this we start by uniformly sampling a weight matrix representing the weighting of connections between nodes in the graph where disconnected nodes have zero connection strength by default. In order to ensure the echo state property is satisfied we scale our sampled weights matrix using the spectral radius $\rho$, to create a weight matrix with unitary spectral radius. The resulting matrix is scaled once more this time by a hyperparameter $S < 1$ to yield the dynamical system weights $\mathbf{W}_{\text{dynamics}}$. This procedure for generating the graph representing our discrete-time nonlinear dynamics is formally outlined below:

$$
\begin{aligned}
A_{ij} &= \begin{cases} 1 & \text{with probability } 0.01 \\ 0 & \text{with probability } 0.99 \end{cases} \\
W_{ij} &= \begin{cases} w_{ij} & \text{if } A_{ij} = 1; w_{ij} \sim \mathbb{U}[-1,1] \\ 0 & \text{if } A_{ij} = 0 \end{cases} \\
\rho(\mathbf{W}) &:= \max\left\{ |\lambda_i| : \lambda_i \text{ is an eigenvalue of } \mathbf{W} \right\} \\
\mathbf{W}_{\text{dynamics}} &= \rho(\mathbf{W})\mathbf{W}S
\end{aligned}
\tag{5}
$$

The discrete-time dynamics or node update rules we define incorporate learnable embeddings using an additive relationship between a learnt and fixed embedding vector as outlined below:

$$
\begin{aligned}
\mathbf{I}(t) &:= \mathbf{W}_{\text{in}}\mathbf{u}(t) + f_{\theta_{\text{in}}}(\mathbf{u}(t), \mathcal{T}) \\
\tilde{\mathbf{x}}(t) &:= tanh(\mathbf{I}(t) + \mathbf{W}_{\text{dynamics}}\mathbf{x}(t-1)) \\
\mathbf{x}(t) &:= (1-\alpha)\mathbf{x}(t-1) + \alpha\tilde{\mathbf{x}}(t)
\end{aligned}
\tag{6}
$$

where $f_{\theta_{\text{in}}}$ represents a learnable function mapping that generates a learnt embedding from input data $\mathbf{u}(t)$ and task relevant data $\mathcal{T}$, while $I(t)$ defines the combination of input embeddings generated by the learnable and fixed input mappings respectively. The remaining terms inherit from Eqn 3 outlined in previous sections of this report. Importantly, this update rule ensures the nonlinear dynamics model leverages learnable embeddings making it possible to incorporate the model into existing neural network architectures and to apply task conditioning as shown in Fig 1.

### 3.1.2 DYNAMICS MODEL INPUT TRANSFORMATION

Our dynamics model requires a fixed and learnable transformation of the input data. The fixed transformation follows from Jaeger & Haas (2004) in that we sample weights for a linear transformation $\mathbf{W}_{\text{in}}$ uniformly at random. The learnable transformation $f_{\theta_{\text{in}}}$ is parameterised by a neural network. This learnable transformation can be used to condition the input state of the dynamics model on task relevant data as shown in Fig 1.

### 3.1.3 DYNAMICS MODEL OUTPUT TRANSFORMATION

In contrast to the canonical linear output mapping of an ESN, outlined in Eqn 4, the output transformation $g_{\theta_{\text{out}}}$ from dynamical system state $x(t)$ to prediction $y(t)$ is represented neural network.

$$\mathbf{y}(t) = g_{\theta_{\text{out}}}(\mathbf{x}(t)) \tag{7}$$

## 3.2 MODEL OPTIMISATION

### 3.2.1 DYNAMICAL SYSTEM PARAMETERS

The optimisation of parameters associated with our dynamics model is mostly addressed as a hyperparameter search problem. In the proposed neural network layer we seek to optimise our model dynamics while preserving the echo state property. In order to accomplish this we employ the procedure that has been outlined in Sec 3.1.1 for ensuring the echo state property is satisfied. In order to search for optimal dynamics parameters we bound the range of parameters to ensure the echo state property and then perform hyperparameter search. The parameters we optimise include (a) the leak rate $\alpha$, (b) spectral radius $\rho$ and (c) scale of weights in the input projection $\mathbf{W}_{\text{in}}$.

### 3.2.2 NEURAL NETWORK PARAMETERS

Learnable neural network parameters are trained via backpropagation over demonstration sequences as outlined in Alg 1. Unlike purely feedforward architectures; architectures that incorporate our ESL layer have predictions that depend on the state $\mathbf{x}_t$ of a nonlinear dynamical system. This has consequences for model training as the model must be trained on sequences of data rather than individual data points. In practice we parallelise the training step across batches of demonstrations and found the learning dynamics and evaluation performance to have greater stability when compared with purely feedforward alternatives.

## 4 EXPERIMENTAL SETUP

### 4.1 LASA HUMAN HANDWRITING DATASET

Our experiments leverage the Human Handwriting Dataset Khansari-Zadeh & Billard (2011) which is composed of demonstrations for 30 distinct human handwriting motions. Each demonstration is recorded on a tablet-PC where a human demonstrator is instructed to draw a given character or shape with guidance on where to start the drawing motion and the point where it should end. For each demonstration, datapoints containing the position, velocity, acceleration, and timestamp are recorded. In our problem formulation, we denote a set of $M$ character demonstrations as $\{\mathcal{D}_j\}_{j=0}^{j=M-1}$. From each demonstration $\mathcal{D}_j$ we select datapoints corresponding to 2D position coordinates $P_j = (p_1, p_2, ..., p_T) : p_k = (x_1, x_2)$ and split the overall trajectory of 2D coordinates into $N$ sequences of non-overlapping windows of fixed length $R$. The resulting dataset is

$\{\{P_{j,i}\}_{i=0}^{N-1}\}_{j=0}^{M-1}$ where each $P_{j,i} = (p_{i*R:(i*R)+R})$ is composed of $N$ non-overlapping subsequences from the 2D position trajectory of the demonstration. Within each subsequence we treat the first datapoint $P_{j,i}^0$ as the current state and model input and the proceedings datapoints $P_{j,i}^{1:R}$ are prediction targets, as a result, all models perform action chunking with a fixed windows size of $R-1$.

## 4.2 BASELINES

### 4.2.1 ECHO STATE NETWORKS

In order to compare our proposed method against current practice with reservoir computing architectures, we implement an echo state network Jaeger & Haas (2004) as one of our baseline models. We employ the same reservoir dynamics as defined in Eqn 3 for this baseline. In order to make a fair comparison with our architecture, we use the same sampling scheme for the fixed weights defining the dynamics; we also ensure the same parameter count and hyperparameter optimisation procedure.

### 4.2.2 ACTION CHUNKING AND TEMPORAL ENSEMBLING

To motivate the efficacy of our approach in overcoming compounding errors compared to existing state-of-the-art feedforward architectures, we baseline against feedforward architectures that incorporate the techniques of action chunking and temporal ensembling introduced by Zhao et al. (2023). When implementing this baseline we assume the same learnable layers as our architecture and remove our neural network layer from the architecture. Similar to other baselines we perform hyperparameter optimisation over the model parameters.

## 4.3 EVALUATION METRICS

### 4.3.1 PRECISION

Assessing the precision of dynamic motions is non-trivial as it involves multiple characteristics including position, velocity, acceleration and jerk. In this work we are primarily concerned with successful character drawing and hence we focus our attention on precision in terms of the ability of the model to reproduce the spatial characteristics of a character. To assess the spatial precision of an architecture we leverage the Fréchet distance metric Eiter & Mannila (1994) between a demonstration curve and the curve produced by the model. While our focus remains on the ability of the model to reproduce the spatial characteristics of a character within a reasonable amount of time, we also recognise the importance of appropriately modelling motion dynamics especially when extending this work to experiments in robotics. As a result we also examine the smoothness of trajectories by evaluating the mean absolute jerk associated with trajectories as well as the Euclidean distance between dynamically time warped velocity profiles for demonstrations and predicted trajectories.

### 4.3.2 GENERALISATION

We are interested in highlighting the flexibility of our approach to adapt to more than one dynamics regime, which to our knowledge hasn't been addressed in traditional reservoir computing literature. To motivate the ability of our approach to generalise we report the average Fréchet distance of a multi-task variant of our architecture given in Fig 1 when trained across multiple character drawing tasks.

### 4.3.3 LATENCY

For each architecture we report the total time taken for the architecture to complete the character drawing task. In practice, the relationship between prediction latency and the dynamics of control is complex, here we aim to highlight the differences in prediction latency alone and reserve the discussion of their interplay for future work.

| | S | | C | | L | |
|---|---|---|---|---|---|---|
| | Fréchet Distance | Latency(ms) | Fréchet Distance | Latency(ms) | Fréchet Distance | Latency(ms) |
| Ours | **2.385** | 3.75 | **1.70** | 3.65 | **1.69** | 3.63 |
| ESN | 4.08 | 4.02 | 3.28 | **3.4** | 1.73 | **3.18** |
| FF | 3.70 | **3.4** | 3.20 | 3.68 | 2.68 | 3.81 |
| FF + Ensemble | 5.02 | 7.21 | 2.92 | 7.02 | 2.40 | 7.06 |

Table 1: The mean Fréchet distance and character drawing latency in milliseconds for all models across individual character drawing tasks "S", "C" and "L".

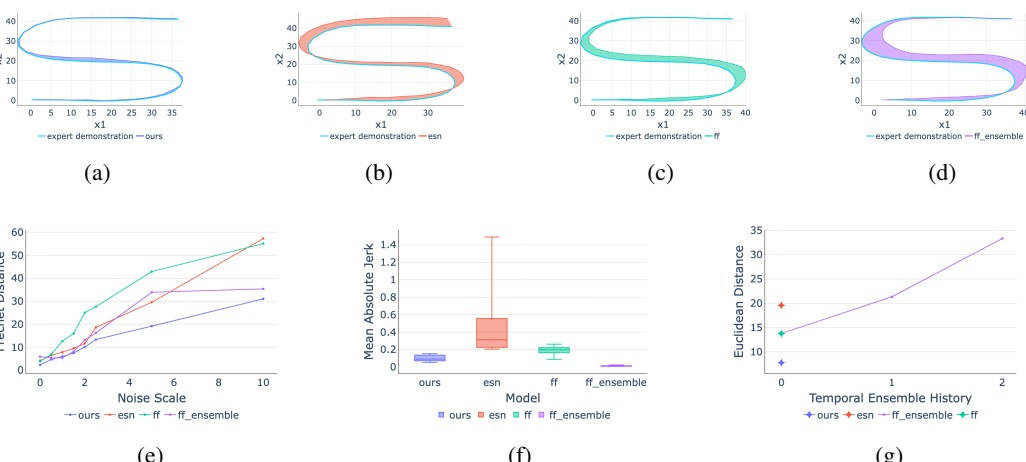

(a)  (b)  (c)  (d)

(e)  (f)  (g)

Figure 3: All plots are for results on the "S" character drawing tasks, similar results are observed for other tasks. **(a-d)** Expert demonstration and predicted drawing trajectories for all models. We include the plots of the area between expert demonstration and predicted trajectories to highlight how well each trajectory aligns with the expert demonstration. **(e)** The Fréchet distance when evaluated for varying levels of random noise, here noise is sampled from a unit Gaussian and scaled according to the noise scale parameter. **(f)** Boxplots of the mean absolute jerk of trajectories generated by the various models. **(g)** Absolute Euclidean distance calculated on dynamically time warped predicted and demonstration trajectories. In this plot, we evaluate the alignment of velocity dynamics for varying levels of temporal ensembling and compare it with our method and the ESN baseline.

## 4.4 EXPERIMENTS

In the following experiments we aim to assess the performance of our architecture under the headings of precision Sec 4.4.1, latency Sec 4.4.2 and generalisation Sec 4.4.3 through benchmarking against the baselines outlined in Sec 4.2.

### 4.4.1 RESULTS ON PRECISION AND OVERCOMING COMPOUNDING ERRORS

In this experiment, we wished to investigate whether our architecture can successfully overcome compounding errors on a given character drawing task by accurately reproducing characters. We report quantitative results on both the single-task Table 1 and multi-task Table 2 settings. In both settings we find that our model consistently obtains the lowest Fréchet distance hence modelling the spatial characteristics of expert demonstrations the best. In general, all models encapsulate the spatial characteristics in the individual character drawing task as seen in the reported Fréchet distances and qualitative plots Fig 5, however, in the multi-task setting this is not always the case with ESNs producing erroneous results in certain cases Fig 7. To further test the ability of each model to overcome compounding errors we introduce random noise at various scales, as demonstrated in Fig 3 **(e)** our model remains the most robust to increasing levels of noise, closely followed by temporal ensembling predictions of feedforward architectures. Unlike our model, temporal ensembling

of predictions comes at the cost of increased latency as seen in Table 1 and a decrease in how accurately task dynamics are modelled. The latter point is highlighted in Fig 3 **(g)** where increasing the history over which ensembling is applied results in increasing disparity between the velocity profile of the expert demonstration and the predicted trajectory.

### 4.4.2 RESULTS ON ARCHITECTURE LATENCY

In this section, we wish to highlight the low computational overhead and the resulting efficacy of our approach in reducing model latency when compared with the technique of temporally ensembling predictions. As outlined in Table 1 and Table 2 temporal ensembling of model parameters results in large increases in the latency of the drawing task (in this case ensembling over history of 2 predictions results in 2-fold increase in latency for completing the drawing). Importantly, ensembling doesn't increase the forward pass prediction latency but rather the number of predictions required to complete the drawing. Without ensembling model predictions all models have similar forward pass latency performance, hence demonstrating that our incorporated neural network layer has minimal computational overhead when compared with an equivalent feedforward architecture without our layer included.

### 4.4.3 RESULTS ON TASK GENERALISATION

|  | Fréchet Distance | Latency (ms) |
| --- | --- | --- |
| Ours | **4.19** | 13.88 |
| ESN | 10.52 | **10.64** |
| FF | 4.62 | 14.16 |
| FF + Ensemble | 4.93 | 25.7 |

Table 2: A comparison of the mean Fréchet distance and latency in milliseconds for all architectures in the multi-task setting.

In this section we aim to highlight how our method builds upon existing reservoir computing literature through demonstrating how task-conditioning with learnable input embeddings enables task generalisation. In Table 2 we see that the ESN baseline demonstrates poor performance in the multi-task setting, with it achieving the largest Fréchet distance metric, which is the result of a failure to converge to the endpoint of a given character drawing. In contrast, our method and feedforward architectures which each incorporate learnable transformation of task relevant data demonstrate the ability to complete the character drawing tasks while maintaining Fréchet distances that are comparable with individual character drawing performance.

## 5 LIMITATIONS AND CONCLUSIONS

In this work, we introduced a recurrent neural network layer for addressing the issue of compounding errors in imitation learning through combining nonlinear dynamics models with the echo state property and traditional neural network components. We demonstrated that the proposed layer improves the performance of existing architectures on the LfD benchmark of learning human handwriting motions from demonstration. In particular, it results in generating trajectories that more closely align with the dynamics of the expert demonstrations while maintaining competitive latency scores. We also validated the integration of our layer into more elaborate architectures of neural network components, demonstrating the ability to combine the generalisation strengths of existing neural network components with our layer to learn to draw multiple characters with one multi-task architecture. A key limitation of architectures incorporating our layer is convergence, notably in the current experiments the model can in certain cases fail to cease making predictions once it has successfully completed a character drawing task which is in contrast to existing work on learning dynamical systems with convergence guarantees, we aim to address this point in future work. Another limitation of the current report is that we have not evaluated the performance of architecture incorporating our layer on real-world tasks involving real hardware, we intend to address this limitation in future work by assessing the performance of the architecture on real-robot manipulation tasks.

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

# A   DYNAMICAL SYSTEM PROPERTIES

Dynamical system properties are a crucial component of our model and the general thesis of this paper; we argue they can be used in the design of temporal inductive biases that are suitable for learning representations. The echo state property of the nonlinear dynamical system we incorporate into our neural network layer is key to the success of our approach. In this section we review the echo state property which is leveraged in our dynamics model and briefly discuss the inductive bias this introduces and how it is well suited to learning representations for modelling temporal data.

## A.1   ECHO STATE PROPERTY

The echo state property defines the asymptotic behaviour of a dynamical system with respect to the inputs driving the system. Given a discrete-time dynamical system $\mathcal{F} : X \times U \to X$, defined on compact sets $X, U$ where $X \subset \mathbb{R}^N$ denotes the state of the system and $U \subset \mathbb{R}^M$ the inputs driving the system; the echo state property is satisfied if the following conditions hold:

$$
\begin{aligned}
&x_k := F(x_{k-1}, u_k), \\
&\forall u^{+\infty} \in U^{+\infty},\ x^{+\infty} \in X^{+\infty},\ y^{+\infty} \in X^{+\infty},\ k \geq 0, \\
&\exists (\delta_k)_{k \geq 0} : ||x_k - y_k|| \leq \delta_k.
\end{aligned}
\tag{8}
$$

where $(\delta_k)_{k \geq 0}$ denotes a null sequence, each $x^{+\infty}, y^{+\infty}$ are compatible with a given $u^{+\infty}$ and right infinite sets are defined as:

$$
\begin{aligned}
U^{+\infty} &:= \{u^{+\infty} = (u_1, u_2, ...) | u_k \in U\ \forall k \geq 1\} \\
X^{+\infty} &:= \{x^{+\infty} = (x_0, x_1, ...) | x_k \in X\ \forall k \geq 0\}
\end{aligned}
\tag{9}
$$

These conditions ensure that the state of the dynamical system is asymptotically driven by the sequence of inputs to the system. This is made clear by the fact that the conditions make no assumptions on the initial state of trajectories, just that they must be compatible with the driving inputs and asymptotically converge to the same state. This dynamical system property is especially useful for learning representations for dynamic behaviours as the current state of the system is driven by the control history.

# B   TRAJECTORY SIMILARITY METRICS

Trajectory similarity metrics are important for assessing how well predicted trajectories align with expert demonstrations, they also give us a sense of the model's ability to overcome the issue of compounding errors when we introduce random noise and perturbations during policy rollouts. In this work, we focus on reproducing the spatial characteristics of expert demonstration trajectories; we choose the Fréchet distance as our evaluation metric since it is well suited to the task of assessing the similarity of human handwriting as demonstrated in existing works in handwriting recognition Sriraghavendra et al. (2007); Zheng et al. (2008). In addition, we assess the smoothness of predicted trajectories through computing the mean absolute jerk of the trajectory. Finally through leveraging dynamic time warping we compute the Euclidean distance between velocity profiles for the predicted trajectory when compared to the expert demonstration.

## B.1   FRÉCHET DISTANCE

Given a metric space $S := (M, d)$, consider curves $C \in S$ of the form $C : [a, b] \to S$ with $a < b$. Given two such curves $P, Q$ of this form, the Fréchet distance between these curves is formally defined as:

$$
F(P, Q) = \inf_{\alpha, \beta} \max_{t \in [0,1]} \{d(P(\alpha(t)), Q(\beta(t)))\}
\tag{10}
$$

where $\alpha, \beta$ take the form of strictly non-decreasing and surjective maps $\gamma : [0, 1] \rightarrow [a, b]$. In our report we leverage an approximation to the Fréchet distance known as the discrete Fréchet distance Eiter & Mannila (1994). In the discrete variant of this metric we consider polygonal curves $C_{\text{poly}} : [0, N] \rightarrow S$ with $N$ vertices which can also be represented as the $(C_{\text{poly}}(0), C_{\text{poly}}(1), ..., C_{\text{poly}}(N))$. For polygonal curves $W, V$ represented by sequences of points, we can consider couplings of points $\mathcal{C}$ from either sequence:

$$(W(0), V(0)), (W(t_{p,1}), V(t_{q,1})), ..., (W(t_{p,M}), V(t_{q,M})), (W(N), V(N)) \tag{11}$$

where the coupling must respect the ordering of points and in either curve and start/end points of the curves are fixed. Given such a coupling the length of the coupling is defined as:

$$||L||_{W,V} := max_{(w,v) \in \mathcal{C}} d((w, v)) \tag{12}$$

where $(w, v)$ is an arbitrary pair from the coupling, this results in the following definition for the discrete Fréchet distance metric:

$$F_{\text{discrete}}(W, V) := \min\{||L||_{W,V}\} \tag{13}$$

## B.2 MEAN SQUARED ABSOLUTE JERK

Jerk is defined as the rate of change of acceleration. Given a discrete set of points $P = (p_0, p_1, ..., p_N)$ representing the positions overtime across an individual dimension of a trajectory (e.g. y coordinates), we can approximate the jerk by taking the third order difference, where importantly we assume a fixed timestep $\Delta t$ between positions:

$$j_i := \Delta^3 p_i = \frac{p_{i+3} - 3p_{i+2} + 3p_{i+1} - p_i}{\Delta t^3} \tag{14}$$

Given a set of jerk values across the trajectory we compute the mean squared absolute jerk as a summary metric as follows:

$$\mathcal{J} := \frac{\sum_{i=0}^{M} |j_i|^2}{M} \tag{15}$$

## B.3 DYNAMIC TIME WARPING

The Dynamic Time Warping (DTW) objective for aligning two time series $X = [x_0, x_2, \ldots, x_{n-1}]$ and $Y = [y_0, y_2, \ldots, y_{m-1}]$ is defined as follows:

$$\text{DTW}(X, Y) = \min_{\pi \in \mathcal{A}(X,Y)} \sum_{(i,j) \in \pi} d(x_i, y_j)$$

where $d(x_i, y_j)$ is a distance metric, $\mathcal{A}(X, Y)$ is the set of all admissible warping paths $\pi$ where a warping path is a sequence of index pairs $\pi = [(i_0, j_0), (i_1, j_1), \ldots, (i_{K-1}, j_{K-1})]$. For a warping path $\pi$ to be admissible, it must satisfy the following conditions:

1. **Boundary Conditions:**

$$(i_0, j_0) = (0, 0) \quad \text{and} \quad (i_K, j_K) = (n - 1, m - 1)$$

This ensures that the warping path starts at the first elements and ends at the last elements of both sequences.

2. **Monotonicity:**

$$i_{k+1} \geq i_k \quad \text{and} \quad j_{k+1} \geq j_k \quad \forall k = 0, 1, \ldots, K-1$$

This condition ensures that the indices in the warping path are non-decreasing, preserving the order of the time series.

3. **Step-size:**

$$(i_{k+1}, j_{k+1}) \in \{(i_k + 1, j_k), (i_k, j_k + 1), (i_k + 1, j_k + 1)\}$$

The path can only move to adjacent timesteps, this ensures that the alignment between the time series is locally contiguous.

## C   ADDITIONAL EXPERIMENTAL DETAILS

### C.1   DATA PREPROCESSING

As outlined in the main text, we denote a set of $M$ character demonstrations as $\{\mathcal{D}_j\}_{j=0}^{j=M-1}$. From each demonstration $\mathcal{D}_j$ we select datapoints corresponding to 2D position coordinates $P_j = (p_1, p_2, ..., p_T) : p_k = (x_1, x_2)$ and split the overall trajectory of 2D coordinates into $N$ sequences of non-overlapping windows of fixed length $R$. The resulting dataset is $\{\{P_{j,i}\}_{i=0}^{N-1}\}_{j=0}^{M-1}$ where each $P_{j,i} = (p_{i*R:(i*R)+R})$ is composed of $N$ non-overlapping subsequences from the 2D position trajectory of the demonstration. Within each subsequence we treat the first datapoint $P_{j,i}^0$ as the current state and model input and the proceedings datapoints $P_{j,i}^{1:R}$ are prediction targets, as a result, all models perform action chunking with a fixed windows size of $R - 1$.

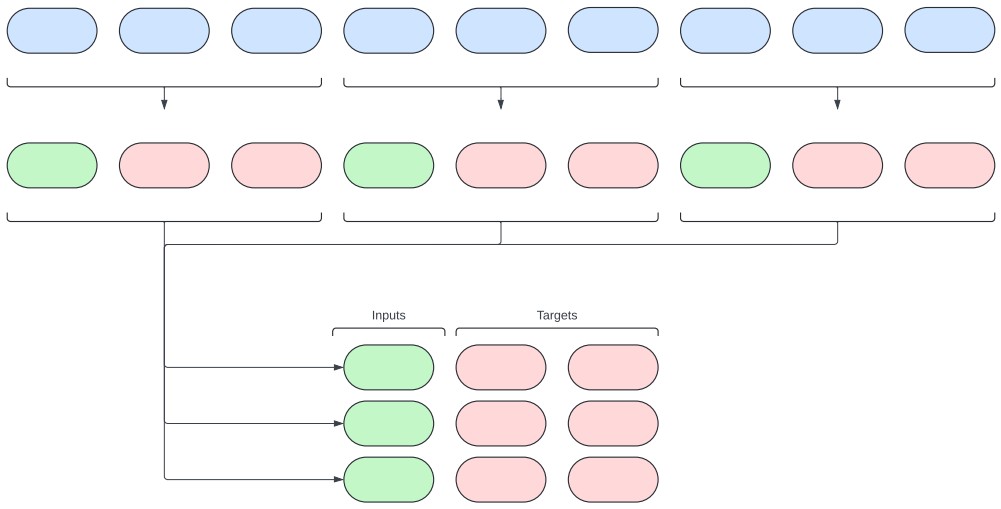

Figure 4: Illustration of taking a trajectory of samples and subsampling it into discrete subsequences (length 3 in the given example) and finally stacking the resulting sequences to create a dataset of inputs and targets.

### C.2   MODEL TRAINING

The proposed neural network layer makes the overall neural network architecture recurrent, therefore, we train the model in a sequential manner as outlined in Alg 1. In practice, for training efficiency we batch sequences of demonstrations, however, for the purpose of outlining the algorithm we demonstrate the individual demonstration case of which the batched case is a generalisation. The convention we adopt for initialising the state of the dynamical system is to initialise it as a vector of zeros $x^T(0) = [0, 0, 0, ..., 0]$.

---

**Algorithm 1** Training Neural Network Parameters - Single-Task Case

---

**Given:** Dataset $\{\{P_{j,i}\}_{i=0}^{N-1}\}_{j=0}^{M-1}$ of subsequences of 2D position trajectories
**Initialize:** $\theta = \{\theta_{\text{in}}, \theta_{\text{out}}\}, W = \{\mathbf{W}_{\text{in}}, \mathbf{W}_{\text{dynamics}}\}, x(0), \alpha$

---

1: **for** $j \leftarrow 0$ to $M - 1$ **do**
2:    **for** $i \leftarrow 0$ to $N - 1$ **do**
3:       $I(i) = \mathbf{W}_{\text{in}} P_{j,i}^0 + f_{\theta_{\text{in}}}(P_{j,i}^0)$
4:       $\tilde{\mathbf{x}}(i) = tanh(I(i) + \mathbf{W}_{\text{dynamics}}\mathbf{x}(i-1))$
5:       $\mathbf{x}(i) = (1-\alpha)\mathbf{x}(i-1) + \alpha\tilde{\mathbf{x}}(i)$
6:       $y(i) = g_{\theta_{\text{out}}}(\mathbf{x}(i))$
7:       $\mathcal{L}(i) = (y(i) - P_{j,i}^{1:R})^2$
8:    **end for**
9:    $\mathcal{L}_{\text{demo}} = \sum_i \mathcal{L}(i)/N$
10:   Update $\theta$ with ADAMW and $\mathcal{L}_{\text{demo}}$
11: **end for**
12: **return** $\theta$

---

### C.3 HYPERPARAMETERS AND HYPERPARAMETER TUNING

The dynamics parameters we optimise include scale of fixed input projection weights, spectral radius, probability of node connections and the leak rate $\alpha$. In the multi-task setting where computational resources limit our ability to run multiple training jobs concurrently, we tune the hyperparameters for the dynamics of our model and the baseline ESN using a Bayesian optimisation implementation provided by the Weights and Biases platform. For single-task training jobs we perform a grid search over candidate hyperparameter sets. This results in the following set of hyperparameters for the dynamics in the multi-task setting:

| | |
|---|---|
| # Nodes | 5000 |
| Node connection probability | 0.01 |
| Input weight range | [-0.1, 0.1] |
| Dynamics weight range | [-0.5, 0.5] |
| Spectral Radius | 0.2 |
| Alpha | 0.25 |

Table 3: Hyperparameters for our model in multi-task setting.

| | |
|---|---|
| # Nodes | 5000 |
| Node connection probability | 0.01 |
| Input weight range | [-0.1, 0.1] |
| Dynamics weight range | [-0.5, 0.5] |
| Spectral Radius | 0.2 |
| Alpha | 0.25 |

Table 4: Hyperparameters for ESN model in multi-task setting.

For training the learnable neural network parameters we use the Adam optimiser with weight decay Loshchilov (2017); Dozat (2016). In order to improve the stability of model training we also normalise gradients exceeding a given threshold Pascanu et al. (2013). In our experiments we found action chunking with $48$ actions to work well and hence each architecture predicts the next 48 actions at every timestep. For optimising the neural network parameters in all architectures we use the following training settings:

| Action Chunks | 48 |
|---|---|
| Weight decay | 0.0001 |
| Gradient Clipping Max Norm | 1.0 |
| Initial learning rate | $1e-7$ |
| Peak learning rate | $1e-3$ |
| Final learning rate | $1e-5$ |
| Learning rate warmup steps | 10 |
| Learning rate decay steps | 2000 |

Table 5: Hyperparameters for neural network parameter training and inference.

# D  QUALITATIVE RESULTS

## D.1  SINGLE-TASK SETTING

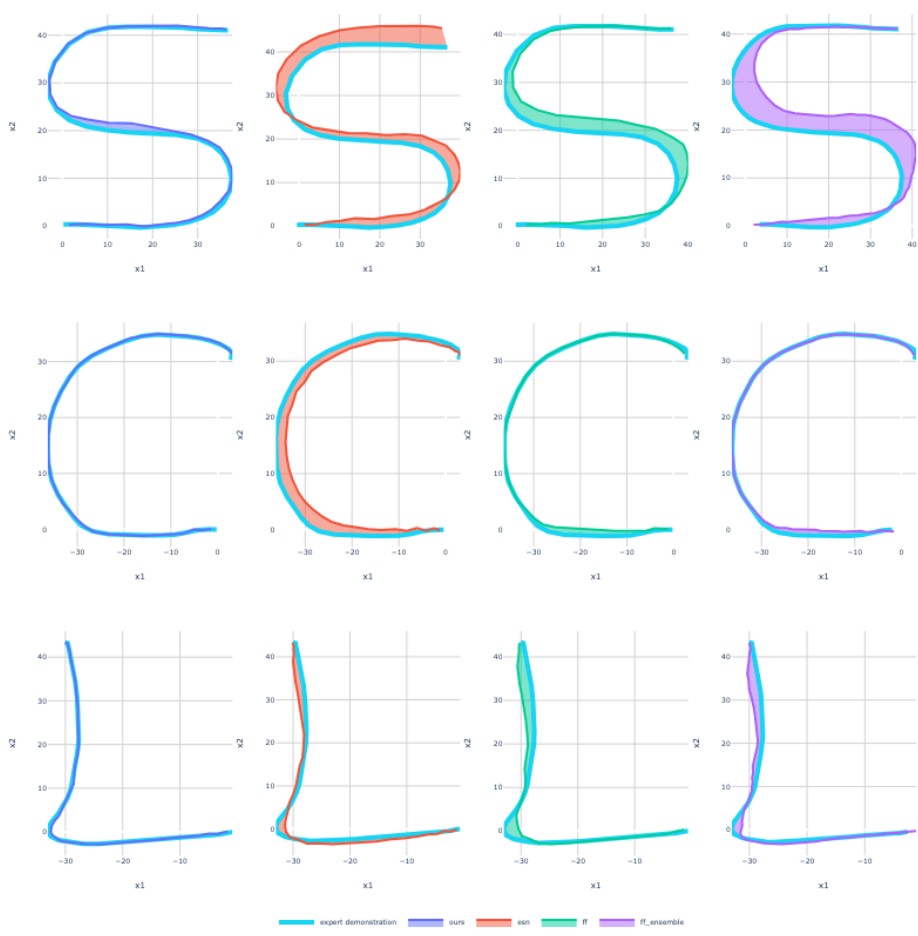

Figure 5: Overlays of an expert demonstration and the trajectory produced by the trained model across the "S", "C" and "L" character drawing tasks for each model.

### D.2 MULTI-TASK SETTING

### D.2.1 OURS

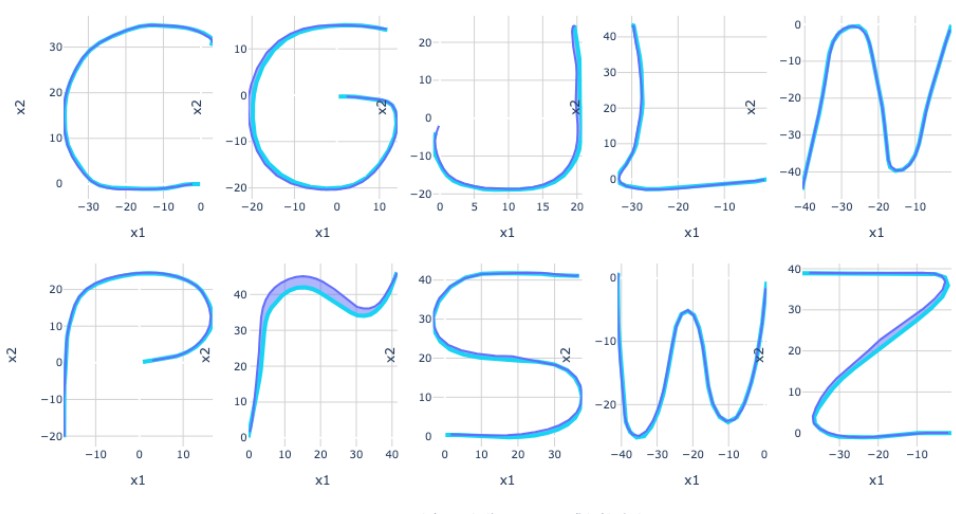

Figure 6: Overlays of an expert demonstration and the trajectory produced by our multi-task model across character drawing tasks.

### D.2.2 ESN

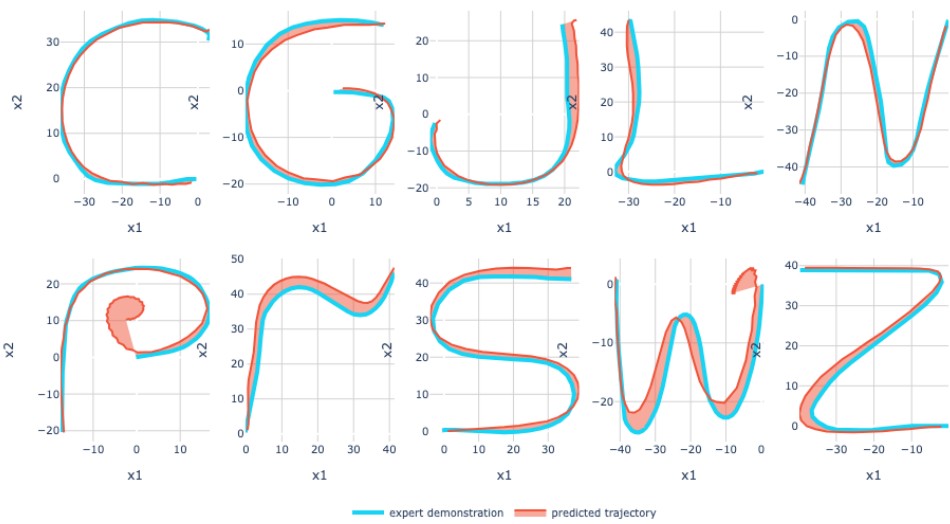

Figure 7: Overlays of an expert demonstration and the trajectory produced by the ESN multi-task model across character drawing tasks.

### D.2.3 FF

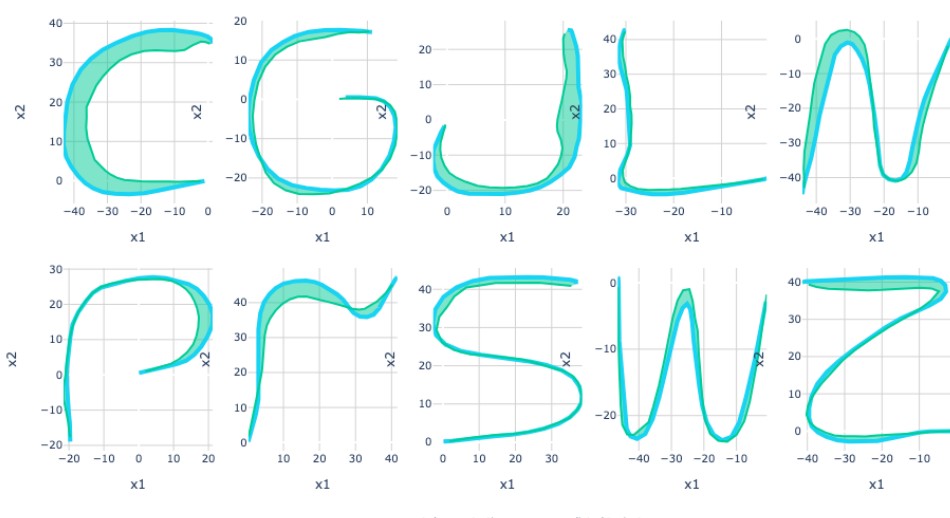

Figure 8: Overlays of an expert demonstration and the trajectory produced by feedforward multi-task model across character drawing tasks.

### D.2.4 FF+ENSEMBLE

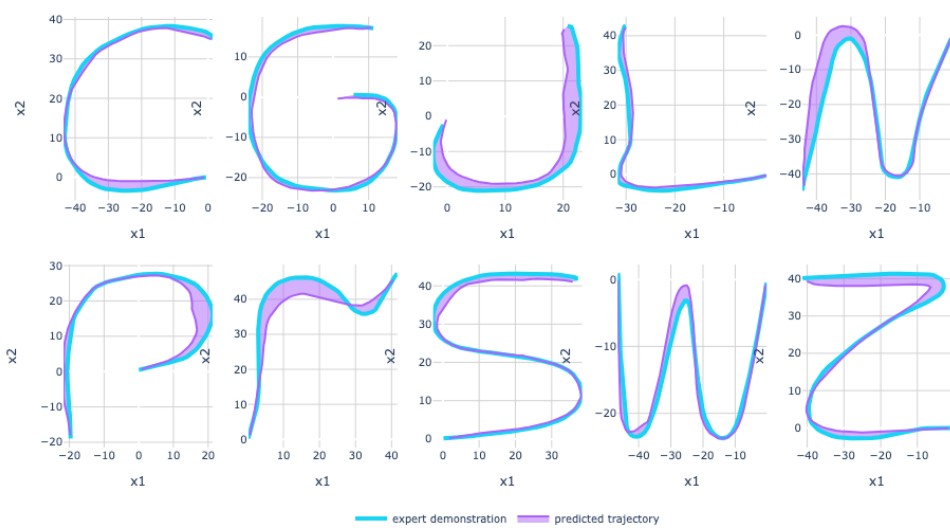

Figure 9: Overlays of an expert demonstration and the trajectory produced by feedforward multi-task model with temporal ensembling across character drawing tasks.

# E    NOTES ON EXTENSIONS

A key motivation of this work is to develop neural network architectures that are well suited to producing precise and dynamic motions on real robot hardware. As a result, we briefly note details of extending this work to real robot hardware. Since our neural network model directly predicts position targets, a low-level controller is required to achieve these targets. To extend this work, we intend to couple our proposed approach to learn a visuomotor policy that relies on images and the current robot state information with an impedance controller used to achieve position targets. The impedance controller ensures compliant motions as the robot interacts with its workspace (subject to hardware constraints). We hypothesise that the strengths of our approach demonstrated in the handwriting task will extend to LfD on real robot hardware resulting in the ability to model a greater set of dynamic behaviours with improved reaction speeds to visual feedback.

In addition, we realise that this paradigm of incorporating dynamical systems into neural network architectures has a lot of potential in real-world robotics tasks. In this paper we argue that this direction should be explored further through optimising the design of the properties of the dynamical systems being used and how they are integrated into the overall architecture. We are excited to continue to pursue this direction and welcome collaborators interested in these topics.

