# OpenReview forum: "Learning from Demonstration with Implicit Nonlinear Dynamics Models"
_ICLR.cc/2025/Conference — Submitted to ICLR 2025_

### Official Review · Reviewer_TFyW · 2024-10-29

**Soundness:** 1
**Presentation:** 2
**Contribution:** 2
**Rating:** 3
**Confidence:** 4

**Summary:**

The paper proposes to use echo state network as layers to fit temporal trajectories in learning from demonstration tasks. The idea includes learning selected parameters of the dynamical system and also a conditional model for contextual input in fitting multi-mode demonstrations. The proposed method is validated in a handwriting dataset and a few metrics on trajectory reproduction precision are reported.

**Strengths:**

1. The paper is easy to understand.
2. The paper approached the problem from the dynamical system perspective, which is worth more attentions in the LfD domain.

**Weaknesses:**

1. The paper only reports results on a handwriting dataset with low-dimension states, lacking comparison to modern LfD approaches in more realistic scenarios, such as robotics.
2. The novelty is unclear. The core technique appears to be completely borrowed from echo state network and the significance of adopting this design in the target context is not sufficiently addressed.
3. The paper motivates with addressing compounding errors in deploying behaviour cloned policies. But this is not demonstrated in the methodology or the experiment sections. The current results and presentation are not coherent with the paper theme.

**Questions:**

1. What makes the proposed layer differ from a recurrent neural network layer? How is this less susceptible to compounding errors as the dynamical system is also time-discretised and explicitly integrated?
2. Will the approach scale up to more complicated motion/policy? How about high-dimension state space beyond 2D? How will it be compared to state-of-the-art methods such as diffusion policy?
3. Even focusing on the handwriting task, can the method be comparable/outperforming classic network approaches like Alex Graves' work?
 4. What is the advantage of the proposed approach considering all existing LfD and handwriting generation models, since none of them are involved in the comparative study?

---

> ### Author Response · Authors · 2024-11-13
> **Initial Review Response**
>
> Thank you for taking the time to review my paper and provide helpful feedback.
>
> "The paper only reports results on a handwriting dataset with low-dimension states, lacking comparison to modern LfD approaches in more realistic scenarios, such as robotics."
>
> I agree, would adding LFD on simulated motions of the Franka Emika Panda using MuJoCo suffice to address this concern?
>
> "The novelty is unclear. The core technique appears to be completely borrowed from echo state network and the significance of adopting this design in the target context is not sufficiently addressed."
>
> The echo state dynamics are certainly not novel but the integration of these dynamics into neural network architectures is novel to the best of my knowledge. I don't disagree with this second point, I can look to include more extensive evaluations to help address this concern. I tried to demonstrate how the proposed approach leads to improved modelling of the demonstration data (Frechet distance) while remaining robust to perturbations and other factors that could cause errors to compound. Do you have suggestions for what you would like to see to sufficiently address this concern, I would be happy to work on this during the rebuttal period.
>
> "The paper motivates with addressing compounding errors in deploying behaviour cloned policies. But this is not demonstrated in the methodology or the experiment sections. The current results and presentation are not coherent with the paper theme."
>
> In the current set of evaluations the perturbations applied during the rollouts are intended to push the policy into out of distribution states that could result in compounding errors, the robustness to perturbations in the paper is intended to assess the ability to counteract compounding errors. This fact becomes even more salient as we reduce the action chunk size. Is there a particular evaluation you would like to see to highlight the ability to overcome compounding errors?
>
> "What makes the proposed layer differ from a recurrent neural network layer? How is this less susceptible to compounding errors as the dynamical system is also time-discretised and explicitly integrated?"
>
> This is a really good question, the core difference is that in a traditional RNN the hidden state is the result of a learnt transformation defined by a set of neural network parameters. In the case of the proposed layer the hidden state results from dynamics with a fixed property (echo state property), this is the core difference. In addition we still have a learnable transformation that operates over the generated hidden states  so BPTT is used here as well just we enforce a different type of inductive bias.
>
> The proposed approach is less susceptible to compounding errors based on how hidden states are updated. The update rule essentially performs exponential smoothing which prevents very large updates that may exacerbate compounding errors. This smoothing also has effects on the task dynamics.
>
> "Will the approach scale up to more complicated motion/policy? How about high-dimension state space beyond 2D? How will it be compared to state-of-the-art methods such as diffusion policy?"
>
> This is not addressed in the current submission but it is very relevant so thanks for brining this up. I suspect it scales to high-dimensional state spaces, I can look to provide an evaluation to demonstrate this point. I would say that this approach can be combined with diffusion policies, the proposed layer is used to generate representations that may be suitable for learning dynamics, I suspect it may provide additional benefits when combined with SOTA such as having a  prediction head that models a denoising diffusion process (or related approach).
>
> "Even focusing on the handwriting task, can the method be comparable/outperforming classic network approaches like Alex Graves' work?"
>
> This is a valid question, in all honesty I didn't have the time to implement more extensive sets of baselines, in addition Alex Graves' work leverages another dataset and is quite extensive to reimplement. I can add a LSTM baseline in order to have a comparison with more classic approaches. Please let me know if this is sufficient to address this concern and if not what you would like to see?
>
> "What is the advantage of the proposed approach considering all existing LfD and handwriting generation models, since none of them are involved in the comparative study?"
>
> The proposed approach instigates a discussion on the use of dynamical system properties for generating inductive biases that are suitable for learning under the LFD paradigm. This work is intended to be shared with the representation learning community with this purpose, it may also have some immediate practical benefits.
>
> Thank you for the valuable feedback, I will look to address your questions with updates to the paper evaluations and codebase.

---

> > ### Comment · Reviewer_TFyW · 2024-11-18
> >
> > Thanks for the authors' rebuttal. The concrete tasks and baselines that may help to prove the generality and soundness of the paper's argument could be referred modern imitation learning literature with high-dim input/output space and execution subject to real system noise. For the example of diffusion policies, one may refer to some RSS 2024 papers [1][2][3].
> >
> > The rate remains unchanged as there are many concerns to address.
> >
> > [1] Zhang et al, Diffusion Meets DAgger: Supercharging Eye-in-hand Imitation Learning, RSS 2024
> > [2] Ze et al, 3D Diffusion Policy: Generalizable Visuomotor Policy Learning via Simple 3D Representations, RSS 2024
> > [3] Prasad et al, Consistency Policy Accelerated Visuomotor Policies via Consistency Distillation, RSS 2024

---

### Official Review · Reviewer_Zkz2 · 2024-10-30

**Soundness:** 3
**Presentation:** 3
**Contribution:** 2
**Rating:** 6
**Confidence:** 2

**Summary:**

This paper develops a recurrent neural network layer that includes a fixed nonlinear dynamical system with tunable dynamical properties for modeling temporal dynamics. And their method outperforms existing approaches including a temporal ensemble of policy predictions and an Echo State Network (ESN) implementation.

**Strengths:**

This paper is well-written, and has clear figures.

The method is introduced in a reasonable and theatrical way.

The results show that their method performs well practically.

**Weaknesses:**

N/A (I'm not an expert in this area, but I'd be happy to get input from other reviewers)

**Questions:**

N/A

---

> ### Author Response · Authors · 2024-11-13
> **Initial Review Response**
>
> Thank you for taking the time to review my paper.

---

### Official Review · Reviewer_6L3F · 2024-11-03

**Soundness:** 2
**Presentation:** 3
**Contribution:** 3
**Rating:** 5
**Confidence:** 2

**Summary:**

This paper proposes a combination of neural networks and nonlinear dynamics models to model sequential data. In comparison to traditional reservoir computing, the additional neural components in the dynamic model allow for better generalization. The authors demonstrate that the new proposed layer reduces the compounding error in LfD on the Human Handwriting Dataset.

**Strengths:**

* A simple idea that updates the ESN with newer deep learning-based components.
* The paper is well-written and presents its ideas clearly, making it accessible and easy to follow.
* Code release for simple integration of the proposed Echo State Layer

**Weaknesses:**

* The method is only evaluated on a single dataset.
* It is challenging to assess its real-world relevance based on the presented experiments. For example, in robotic manipulation tasks, the practical benefits of this approach remain uncertain.

**Questions:**

* Would it be possible to show the layer's effectiveness on tasks closer to robotics, such as the MIME Dataset?
* Or on datasets with larger input/output dimensions?
* Is there a limit in the dimensionality where one baseline would start to get the upper hand? A simple experiment with the handwriting dataset would be to treat multiple characters together as a single character in a higher-dimensional space. E.g., two characters are then represented as one character with a 4D curve (u1, u2, u3, u4)
* How does the proposed architecture compare to deep learning methods that use Behavioral Cloning or Dagger-like approaches? Is there a way to use ideas from these algorithms to train the ESL?

---

> ### Author Response · Authors · 2024-11-13
> **Initial Review Response**
>
> Thank you for taking the time to review my paper and provide helpful feedback.
>
> "Would it be possible to show the layer's effectiveness on tasks closer to robotics, such as the MIME Dataset?
>
> Or on datasets with larger input/output dimensions?"
>
> Yes I will look to add evaluations of motions in a MuJoCo simulated environment. This is very relevant and something I wanted to get complete but I didn't before the submission date.
>
> "Is there a limit in the dimensionality where one baseline would start to get the upper hand? A simple experiment with the handwriting dataset would be to treat multiple characters together as a single character in a higher-dimensional space. E.g., two characters are then represented as one character with a 4D curve (u1, u2, u3, u4)"
>
> This is interesting question, I can't answer this definitively yet without empirical evidence. I suspect the results would largely remain the same taking into account any necessary parameter tuning that is required for different task definitions.
>
> "How does the proposed architecture compare to deep learning methods that use Behavioral Cloning or Dagger-like approaches? Is there a way to use ideas from these algorithms to train the ESL?"
>
> The current training scheme relies on behaviour cloning, the input and output transformations are represented by learnable neural network parameters that are trained via backpropagation through time. DAgger could certainly be used to improve the performance through collecting further data for training.
>
> Thank you for the valuable feedback, I will look to follow up in revisions.

---

### Official Review · Reviewer_iQBH · 2024-11-04

**Soundness:** 3
**Presentation:** 3
**Contribution:** 2
**Rating:** 3
**Confidence:** 4

**Summary:**

The paper proposes a method for incorporating non-linear dynamic systems in a policy representation for learning from demonstration (LfD). The approach extends the echo state networks architecture to include learned components to embed the inputs of the policy (pen state + image to be drawn). These input embeddings are then integrated into the architecture in a way that influences the dynamics of the layer. The work is evaluated on the LASA human handwriting dataset. The baselines are: feedforward networks, echo state machine, and temporal ensembling. The results show that the proposed method improves precision and generalization without strongly compromising the jerkiness of the movements and the latency of the inference.

**Strengths:**

- the paper is well-written and enjoyable to read.
- the paper is self-contained and includes most of the needed background knowledge for an author to follow.
- the proposed method is quite interesting. It makes a lot of sense to model temporal dynamics with non-linear dynamical systems for learning from demonstration.
- the experiments study multiple metrics that are relevant to the LASA dataset.
- the results are very promising and nicely demonstrate the benefits of the method, namely improving precision and generalization without strongly compromising latency.

**Weaknesses:**

My main concern with this work is in its evaluations:
- the experiments do not include multiple baselines that account for context or memory such as transformer, SSMs, LSTMs...
- the experiments are limited to a single small-scale dataset. It would be interesting to understand how the proposed method would perform on various LfD tasks. Ideally, it would be nice to include tasks that require the policy to be reactive, for instance, tasks involving interaction with an object like pushing, or sticking to your example, drawing the drawings from different starting positions.
- the paper lacks ablations of some components and hyperparameters it includes (ResNet, $\alpha$...).

**Minor issues:**

- punctuation of equations is missing for most equations.
- line 341 --> represented using/by a neural network.

I am willing to raise the score (to a positive score) if more baselines and tasks are included in the evaluation.

**Questions:**

- why does the proposed approach lead to a higher jerk than the temporal ensembling baseline (figure 3f)?
- why the latency of the ESN is worse than their method in Table 1 but better in Table 2?
- can you elaborate further on how the proposed method ensures the preservation of the echo state property?
- how would the proposed method perform on more reactive and complex tasks, for instance, tasks involving interaction with an object like pushing, or sticking to your example, drawing the drawings from different starting positions?
- how does the method compare to transformer, SSM, and LSTM-based architectures?
- there are multiple ways of implementing temporal ensembling, can you elaborate on your temporal ensembling approach?

For all questions, please provide possible explanations or hypotheses where suited.

---

> ### Author Response · Authors · 2024-11-13
> **Initial Review Response**
>
> Thank you for taking the time to review my paper and provide helpful feedback.
>
> "the experiments do not include multiple baselines that account for context or memory such as transformer, SSMs, LSTMs..."
>
> This is a very valid point, I will look to add a LSTM and transformer baseline to address this inherent weakness.
>
> "the experiments are limited to a single small-scale dataset. It would be interesting to understand how the proposed method would perform on various LfD tasks. Ideally, it would be nice to include tasks that require the policy to be reactive, for instance, tasks involving interaction with an object like pushing, or sticking to your example, drawing the drawings from different starting positions."
>
> I can look to extend the evaluation to simulated robotics motions using Franka Emika Panda in MuJoCo. For the reactivity point, the current method will definitely require tuning of the state update step, as this is essentially exponential smoothing whose parameters impact the responsiveness of motions to new inputs. It is worth mentioning that this is dependent on the relationship between prediction latency and underlying controller frequency. One option is to make the update parameters tunable based on inputs which should help address this problem but I will need to implement this extension. To address this point I'll look to simulate a toy task in MuJoCo that requires responsiveness to visual inputs.
>
> "the paper lacks ablations of some components and hyperparameters it includes (ResNet, ...)."
>
> I'll try to add more ablations, I definitely agree with this point.
>
> "I am willing to raise the score (to a positive score) if more baselines and tasks are included in the evaluation."
>
> Thanks, this work is very much an initial step in the direction of developing architectures that incorporate ideas from dynamical systems theory (for which there is much existing work). I do think the current work is far from perfect, however, I think it has the potential to instigate more discussion around this general direction and it may be practically useful but there is certainly more work required to further develop related approaches.
>
> "why does the proposed approach lead to a higher jerk than the temporal ensembling baseline (figure 3f)?"
>
> This is an excellent question and something I was trying to understand when preparing the results as it is very relevant for real robot experiments. I think this result depends very much on the parameters for both the proposed layer and temporal ensembling. In the proposed layer, the parameter alpha weighs the state update step and was set to 0.25, in contrast the temporal ensembling weight parameter was set to 0.001 with a prediction horizon of 2. The relationship between these parameter choices and the resulting jerk in motions would address your question. I didn't include this in the paper but I think it may be feasible to draw a direct comparison between the weighting applied in both approaches, the current discrepancy is likely an artefact of the parameters used.
>
> "why the latency of the ESN is worse than their method in Table 1 but better in Table 2?"
>
> Yeah this is a confusing result, I wanted to reformat this table to be more informative. This metric is suboptimal as it is a combination of the mean prediction latency times number of predictions required to complete the drawing task. The short answer is in the difference in the number of predictions between the single and multi-task case.
>
> In the single task case all architectures make essentially the same number of predictions and hence the latency score is more indicative of computational overhead. Here the ESN generally performs best, it is only on the S character task it performs worse. In the multi-task case, the ESN solves the task in less predictions but it is highly inaccurate on fitting the spatial characteristics of the demonstration and hence while it converges to the expected end point for the task in less predictions it doesn't fit the data very well. This isn't communicated well in this table so I'll look to add information on the number of predictions such that it communicates the prediction latency and number of predictions.
>
> "can you elaborate further on how the proposed method ensures the preservation of the echo state property?"
>
> So in the shared code there is actually a typo that needs to be addressed. The echo state property is actually maintained according to the following scheme: https://www.sciencedirect.com/science/article/abs/pii/S0893608012001852?via%3Dihub
>
> "how would the proposed method perform on more reactive and complex tasks, for instance, tasks involving interaction with an object like pushing, or sticking to your example, drawing the drawings from different starting positions?"
>
> This still needs to be tested but likely it would work well with the appropriate parameter tuning.
>
> ** I am running out of space so I will follow up on the other questions in another official comment.

---

> > ### Author Response · Authors · 2024-11-13
> > **Initial Review Response - CTD**
> >
> > "how does the method compare to transformer, SSM, and LSTM-based architectures?"
> >
> > This is a very valid question I will need to add baselines for these architectures to address this question definitively.
> >
> > "there are multiple ways of implementing temporal ensembling, can you elaborate on your temporal ensembling approach?"
> >
> > The temporal ensembling approach follows directly from https://arxiv.org/pdf/2304.13705.
> >
> > Thank you for the valuable feedback, I will look to address your questions with updates to the paper evaluations and codebase.

---

### Meta-Review · Area_Chair_Ur36 · 2024-12-24

**Metareview:**

This paper proposes a method that integrates fixed non linear dynamical systems with learned embeddings to address compounding errors in learning from demonstrations. Experiments on the LASA dataset shows improved spatial performance, robustness to noise and lower latency compared to feed forward networks and echo state nets.

The writing is relatively clear. The hybrid approach of combining reservoir computing ideas with learned embedding transformations is an interesting contribution. The authors’ experiments demonstrate improved precision and robustness on a benchmark handwriting dataset. They also provide an open-source library, potentially useful for further research.

The evaluation focuses on a single low-dimensional dataset. Reviewers requested more comprehensive baselines (e.g., LSTMs, Transformers, Diffusion Policies) and tests in real or more complex simulation scenarios. Additional ablations and deeper empirical evidence of handling compounding errors in realistic tasks would strengthen the claims of the paper.

**Additional Comments On Reviewer Discussion:**

During discussion, reviewers noted the limited experimental scope (only handwriting) and the lack of mainstream sequence-modeling baselines. They also highlighted the need for more complex tasks. The authors plan to add more baselines and experiments (e.g., in MuJoCo) to address these concerns. However, reviewers felt these additions were necessary before acceptance.

---

### Decision · Program_Chairs · 2025-01-22

Reject